# Effects of Whole-Body Electromyostimulation on Metabolic Syndrome in Adults at Moderate-to-High Cardiometabolic Risk—A Systematic Review and Meta-Analysis

**DOI:** 10.3390/s24216788

**Published:** 2024-10-22

**Authors:** Ellen Guretzki, Matthias Kohl, Simon von Stengel, Michael Uder, Wolfgang Kemmler

**Affiliations:** 1Institute of Radiology, University Hospital Erlangen, Henkestrasse 91, 91052 Erlangen, Germany; e.guretzki@htp-tel.de (E.G.); simon.von.stengel@fau.de (S.v.S.); michael.uder@uk-erlangen.de (M.U.); 2Department of Medical and Life Sciences, University of Furtwangen, 78056 Schwenningen, Germany; matthias.kohl@hfu.eu; 3Institute of Medical Physics, Friedrich-Alexander-University of Erlangen-Nürnberg, 91052 Erlangen, Germany

**Keywords:** whole-body electromyostimulation, electrostimulation, intervention, cardiometabolic risk, metabolic syndrome, obesity

## Abstract

In the present work, we aimed to determine the effect of whole-body electromyostimulation (WB-EMS) on metabolic syndrome (MetS) as a cluster of cardiometabolic risk factors in people at moderate-to-high cardiometabolic risk. The present meta-analysis is based on a systematic literature search of a recent evidence map, which searched five electronic databases, two registers, and Google Scholar, according to PRISMA, until 31 March 2023. Controlled trials comprising adult cohorts with central obesity that compared the effect of WB-EMS versus controls using a continuous score representing MetS were included. We applied a random-effects meta-analysis and used the inverse heterogeneity model to analyze the data of the five eligible trials identified by our search. Outcome measures were standardized mean differences (SMDs) with 95% confidence intervals (95%-CIs). The risk of bias was determined using the PEDro-Score. In summary, we identified five eligible articles containing 117 participants in the WB-EMS group and 117 participants in the control group. We observed a small effect (SMD: −0.30; 95%-CI: −0.04 to −0.56) in favor of the WB-EMS intervention. The heterogeneity between the trials was very low (I^2^: 0%); further evidence for risks of small study/publication bias was minimal. The methodologic quality of these studies can be classified as moderate to high. In summary, the present work provides evidence of the favorable effect of WB-EMS on cardiometabolic risk in adults at moderate–high cardiometabolic risk. Considering the time effectiveness of WB-EMS, along with its safety and attractiveness, as indicated by the five studies, WB-EMS can be regarded as a feasible training option for people at cardiometabolic risk.

## 1. Introduction

Whole-body electrostimulation (WB-EMS) is an increasingly popular innovative training technology. With its ability to stimulate all the main muscle groups simultaneously, but with a dedicated intensity per electrode, WB-EMS can be considered a time-effective, jointly friendly, and safe training method [1,2]. This might qualify this novel training technology as a promising tool to address people with poor health, limited time resources, and/or a low affinity to conventional exercise. Because of the resistance-type character of WB-EMS [3], most studies of sedentary or at least non-athletic cohorts addressed outcomes related to musculoskeletal conditions or diseases [4]. However, some studies provided considerable evidence of the positive effects of a standard WB-EMS application [3] on parameters related to cardiovascular health [5,6,7,8], e.g., in people with diabetes mellitus or chronic heart failure.

Metabolic syndrome (MetS) is a cluster of conditions that increase the risk of heart disease, stroke, type 2 diabetes, and other serious health problems [9]. The components of MetS include (central) obesity, high blood pressure, high levels of triglycerides, low levels of HDL-C, and insulin resistance [9]. Of note, MetS is necessarily dichotomous for clinical purposes, wherein if at least three of five MetS criteria apply, MetS is present [9,10,11]. However, to reliably determine the overall effect of an intervention, a continuous MetS-(Z)-Score is more accurate and appropriate for scientific research [12]. Among others, Johnson et al. [12] suggest calculating a Z-score using individual subject data, MetS cut-off criteria, and standard deviations (denominators of each factor in the formula) of the given (here, female) cohort at baseline (e.g., ([50 − HDL-C]/SD − HDL-C) + ([Triglycerides − 150]/SD − TGs) + ([fasting glucose − 100]/SD − FPG) + ([waist circumference − 88]/SD − WC) + ([mean arterial blood pressure − 100]/SD − MAP)). Johnson et al. [12] applied the National Cholesterol Education Program (NCEP) Expert Panel on Detection, Evaluation, And Treatment of High Blood Cholesterol in Adults (Adult Treatment Panel III). The cut-offs are waist circumference: 102 cm (men) or 88 cm (women), HDL-C: 40 mg/dL (men) or 50 mg/dL (women), triglycerides: 150 mg/dL, fasting glucose: 100 mg/dL, and mean arterial blood pressure (MAP): 100 mmHg.

In the present mini-review and meta-analysis, we aimed to determine the effects of WB-EMS on the MetS-Score. We hypothesized that WB-EMS interventions generated significant effects on the MetS-Score compared with controls in adults at increased cardiometabolic risk.

## 2. Methods

### 2.1. Information Sources and Search Strategy

A literature review was performed to identify the most relevant quantitative and qualitative studies following the Preferred Reporting Items for Systematic Reviews and Meta-Analyses (PRISMA) guidelines.

The present meta-analysis is based on the large systematic literature search and evidence map provided by Beier et al. [1], which was slightly adapted for the present topic. Briefly, publications in five electronic databases (Medline [PubMed], the Cochrane Central Register of Controlled Trials [CENTRAL], the Cumulative Index to Nursing and Allied Health [CINAHL via Ebsco Host], SPORTDiscus (via Ebsco Host), and the Physiotherapy Evidence Database [PEDro]) and two study registers (Clinical trial.gov and the WHO’s International Clinical Trials Registry Platform [ICTRP]) from their initiation to 6 March 2023 were searched without language restrictions. Further, we hand-searched Google Scholar to 6 March 2023. For more detailed information, the reader is referred to the systematic literature search and evidence map created by Beier et al. [1]. In order to ensure that all eligible studies were identified and included in the present analysis, we updated our search (28 September 2024); however, we restricted our search to the electronic databases listed above.

### 2.2. Selection Process

Titles, abstracts, and full texts were independently screened by three reviewers (MB, EG, WK) according to the pre-specified eligibility criteria listed below.

### 2.3. Eligibility Criteria

The eligibility criteria applied to the present systematic review were categorized according to the PICOS scheme.

*Population:* Non-athletic adult cohorts with central (abdominal) obesity according to the cut-off values (men: 94 cm, female: 80 cm) suggested by the International Diabetes Federation (IDF) for the definition of MetS were included.

*Intervention:* We only considered studies that applied whole-body electromyostimulation (WB-EMS) according to the current definition [2].

*Comparators:* All types of control groups, be they physically inactive or active, were considered. Studies with an isolated WB-EMS intervention arm without a control group were excluded. In cases of more than one control group [13], the non-exercise control group was included in the analysis. In cases of superimposed interventions [14], we compared the mixed WB-EMS/exercise group with the isolated exercise group.

*Outcomes:* For the present review, we included eligible studies that reported data on the metabolic syndrome score specified as a primary or secondary outcome. All kinds of continuous scores representing MetS were accepted.

*Study design:* We included only randomized and non-randomized controlled trials. Review articles, case reports, editorials, conference abstracts, letters, or theses (doctoral, master, bachelor) were not considered.

### 2.4. Data Items and the Data Collection Process

A Microsoft Excel table modified for the present research topic was used to extract relevant data from the included studies. Briefly, we extracted publication, study, intervention characteristics, and outcomes. We further recorded whether the MetS-Score was defined (or at least considered) as a primary/main or secondary/subordinate study outcome. Adverse effects related to the WB-EMS intervention were also recorded. Adverse effects were defined as any untoward medical occurrence, unintended disease, or injury or any untoward clinical sign, including an abnormal laboratory finding related to the WB-EMS application.

### 2.5. Risk of Bias Assessment

Risk of bias was classified by WK and SvS using the Physiotherapy Evidence Database (PEDro) Scale Risk of Bias Tool [15] specifically dedicated to physiotherapy/exercise studies and thus appropriate for rating the methodologic quality of the WB-EMS intervention.

### 2.6. Data Synthesis

Missing standard deviations (SDs) were calculated using the method detailed in the recently published comprehensive meta-analysis by Shojaa et al. [16]. If the studies presented a confidence interval (CI) or standard errors (SEs), they were converted to standard deviation (SD) with standardized formulas [17]. Because of the low number of eligible studies, no subgroup analyses were conducted.

### 2.7. Statistical Analysis

A random-effects meta-analysis was computed using the metafor package [18] that is included in the statistical software R [19]. The effect size was presented as standardized mean differences (SMDs) with 95% confidence intervals (95%-CIs). We applied the heterogeneity (IVhet) model proposed by Doi et al. [20]. Heterogeneity between the studies was checked using I^2^ statistics. In addition to funnel plots, regression tests, and rank correlation effect estimates and their standard errors using the *t*-test and Kendall’s τ-statistic for possible small study/publication BIAS, we performed a trim-and-fill analysis using the L0 estimator. In addition, we used DOI plots, the Luis Furuya–Kanamori index (LFK index) [21], regression, and rank correlation tests to check for asymmetry. Sensitivity analyses were applied to determine whether the overall result of the analysis was robust to the use of the imputed correlation coefficient (minimum, mean, or maximum). Further, a sensitivity analysis was applied to determine the potentially confounding effect of a trial with a training control group [22]. SMD values of >0.2, >0.5, and >0.8 were interpreted as small, medium, and large effects. A *p*-value < 0.05 was used as the significance level for all tests.

## 3. Results

### 3.1. Study Selection

Figure 1 (flowchart) illustrates the process of the systematic search conducted in this study. After removing 577 duplicates, 637 articles were screened based on their titles and abstracts. The full texts of 225 potentially relevant articles were screened and, finally, a total of five eligible studies were included [13,14,22,23,24]. After removing duplicates (PubMed: n = 78, CENTRAL: n = 50, CINAHL: n = 18, SPORTDiscus: n = 27, PEDro: n = 0) the updated search to 28 September 2024 identified 80 articles, which were screened for eligibility. However, no further study was included.

### 3.2. Study, Participant, and Exercise Characteristics

The five studies comprised five WB-EMS groups (GC) that were compared with their five most suitable control groups (Table 1). All the studies were randomized controlled trials with parallel group designs that applied balanced randomization. The pooled number of participants (baseline) was n = 117 in the WB-EMS group and n = 117 in the control group. Two trials included only women [23,24], one study focused on men [22], and two studies included mixed cohorts [13,14]. The mean age of the cohorts ranged between 43 ± 6 years [22] and 77 ± 3 years [24] (Table 1). Two studies focused on people with MetS [13,23] according to the definition of the International Diabetes Federation (IDF [9]). One study addressed older women with sarcopenic obesity [24], and two studies included predominately middle-aged participants with overweight and obesity [14,22].

The studies varied considerably with respect to WB-EMS application (Table 2). Most importantly, one study [14] applied superimposed WB-EMS by adding WB-EMS to high-intensity (aerobic and resistance type) interval training (HIIT). All of the other studies focused on WB-EMS with low-intensity voluntary exercises/movements during the WB-EMS impulse phase. Of importance, two studies implemented (conventional) exercise control groups [14,22] (Table 1). While Amaro-Gahete et al. [14] compared WB-EMS and HIIT versus HIIT only, Kemmler et al. [22] compared WB-EMS versus single-set high-intensity resistance exercise (HIT-RT) (Table 1). The length of the interventions varied between 12 and 26 weeks [24], and the average WB-EMS volume ranged between 20 min [24] and 52.5 min/week [14]. Impulse intensity as prescribed by the rate of perceived exertion (RPE) varied from 6 (hard to hard+) [24] to 9 (very, very hard) [14] on the Borg CR10 scale [26].

Loss to follow-up in the WB-EMS groups ranged between 4% for 26 weeks [24] and 25% for 12 weeks of intervention [13]. The withdrawal rate, defined as voluntary drop-out due to personal reasons (e.g., loss of interest, lack of time, aversion to or discomfort with the intervention) and considered an indicator of attractiveness, averaged between 4% and 13%. The attendance rate to the training sessions in the WB-EMS and active control groups averaged ≥ 90%. None of the studies reported any adverse effects related to the WB-EMS or exercise interventions.

Energy-restrictive diets were applied in two of the five studies [13,23]. Reljic et al. reported a net energy reduction of −502 (WB-EMS) vs. −439 kcal/d (CG) [13] and −336 (WB-EMS) versus −588 kcal (CG) [23], in their groups. However, the authors did not list significant group effects. Three studies monitored dietary habits in their WB-EMS and CG-groups [14,22,24,27]. While Amaro-Gahete et al. [14] listed marginal changes in energy intake in HIIT and HIIT + WB-EMS (36 vs. 11 kcal/d), Kemmler et al. [22] observed a significant difference (2.9 ± 9.9% vs. 7.8 ± 10.6%, *p* = 0.010) between the groups with higher intake in the WB-EMS-group. Lastly, Wittmann et al. [24] reported significant reductions (−139 kcal/d, *p* = 0.019) only for the WB-EMS group, whilst significant differences to the CG were not reported.

### 3.3. Methodologic Quality of the Trials

Following PEDro and applying the classification of Ribeiro de Avila et al. [28], the methodologic quality of the studies can be classified as moderate (PEDro: 5–7) to high (PEDro: ≥8) (Table 3). In particular, aspects related to allocation concealment or blinding prevented better ratings.

### 3.4. Study Outcomes

All trials [13,14,22,23,24] treated the MetS-Z-score (…or the “cardiometabolic risk profile”) as the primary or main study outcome. However, two [13,23] studies applied the MetS-Syndrome definition specified by the National Cholesterol Education Program (NCEP) Adult Treatment Panel III [10], and three other studies applied the IDF criteria and cut-off values for the metabolic syndrome [14,22,24]. Briefly, both definitions used the same cut-off values for HDL-C (men: 40 mg/dL, women: 50 mg/dL), triglycerides (150 mg/dL), and fasting glucose (100 mg/dL); however, different values for blood pressure (IDF: diastolic 85 mmHg, systolic: 135 mmHg versus NCEP-ATP III: mean arterial blood pressure (MAP): 100 mmHg) and, in particular, waist circumference (IDF: men 94 cm, women: 80 cm versus NCEP-ATP III: men: 102 cm, women: 88 cm) were used.

As stated, all the studies summarized the five MetS components as a continuous score. Apart from one study [14] that did not provide sufficient information, all the trials calculated MetS-Z-scores according to the approach suggested by Johnson et al. [12]. Applying the NCEP ATP III cut-off values for a female cohort, the MetS-Z-score was calculated as follows: MetS-Z-score: ([50 − HDL-C]/SD − HDL-C) + ([Triglycerides − 150]/SD − TGs) + ([fasting glucose − 100]/SD − FPG) + ([waist circumference − 88]/SD − WC) + ([mean arterial blood pressure − 100]/SD − MAP). Amaro-Gahete et al. [14] applied quite a similar approach. Briefly, the authors divided the sum of the five (waist circumference + MAP + glucose + triglycerides + HDL-C) standardized scores (value–mean/SD) to calculate a continuous score.

Relevant for the interpretation of the results, decreases in the MetS-Score(s) always indicate favorable changes.

### 3.5. Meta-Analysis Results

Figure 2 displays the results of WB-EMS versus control on the MetS-Score. In summary, we observed a low (SMD: −0.33; 95%-CI: −0.07 to −0.59) but significant (*p* = 0.013) effect in favor of the WB-EMS intervention. Heterogeneity between the trials was very low (I^2^: 0%, Figure 2). In the sensitivity analysis with respect to the imputation of the mean correlation (see Figure 2), the minimum or maximum correlation revealed roughly comparable effects. Of note, the two studies that compared WB-EMS with an exercise control group did not negatively impact the result of the meta-analysis (Figure 2).

Excluding the trial by Kemmler et al. [22], which compared WB-EMS with HIT-RT, slightly reduced the effect of WB-EMS on the MetS-Z-score (SMD: 0.29, 95%-CI: −0.00 to −0.58), but it was still significant (*p* = 0.049).

The IV-Het funnel plot with the trim-and-fill analysis (Figure 3) imputed one study at the lower right side, thus indicating a publication/small study bias. The LFK Index (1.61) indicated minimal asymmetry; in parallel, the regression (*p* = 0.526) and rank correlation test (*p* = 0.817) did not indicate significant asymmetry.

Table 4 lists changes in the components of the MetS-Z-score in detail. In summary, only MAP revealed consistently more favorable results in the WB-EMS group compared with the control group, although only one study stated significant differences [24]. However, only three of the five studies applied statistical tests to address this issue.

## 4. Discussion

The present systematic review and meta-analysis, including five randomized controlled trials that considered the MetS-Score as the main outcome, shows a low positive effect (SMD: −0.30; 95%-CI: −0.04 to −0.56) of WB-EMS application on this cardiometabolic risk cluster in middle-aged to older women and men with increased cardiometabolic risk. This significant finding is quite impressive since two trials [14,22] implemented active control groups with exercise protocols (HIIT and HIT-RT), with high evidence of positive effects on cardiometabolic risk factors related to MetS [29,30,31]. Amaro-Gahete et al. [14] compared combined WB-EMS/HIIT versus isolated HIIT and reported significant positive changes for the combined group only without any change in HIIT. Of note, the non-training control group in their study deteriorated significantly; thus, significant effects (compared with the CG) were reported for HIIT and WB-EMS/HIIT. In parallel, (significantly) unfavorable changes in the CG with significant effects were also reported for the homeostasis model assessment index (HOMA) and the quantitative insulin sensitivity check index (QUICKI). Nevertheless, this significant deterioration in important cardiometabolic biomarkers within 12 weeks in the CG that received no intervention within that period is surprising, particularly since no confounding effects were reported.

Kemmler et al. [22] compared MetS-Z-score changes after 16 weeks of WB-EMS versus a comparable time-effective HIT-RT (i.e., single-set RT with high exercise intensity). The protocol reported non-significantly (*p* = 0.096) more favorable results for WB-EMS. In contrast, after 12 weeks of intervention, Reljic et al. [13] reported the opposite effects after comparing their single-set RT group (not included in the analysis) with WB-EMS. The longer intervention period and higher exercise intensity in the study by Kemmler et al. [22] may well contribute to this finding. In addition, Reljic et al. [13] also reported significantly more favorable effects after HIIT and multiple-set RT (not included in the analysis) compared with WB-EMS. However, given the proof-of-principle approach of the present study, we decided not to compare conventional exercise versus WB-EMS at least when non-training control groups were available. Since only one study was included, by directly comparing the effects of WB-EMS versus HIT-RT [22,27], we are unable to reliably decide whether DRT or WB-EMS is superior for favorably affecting the MetS-Z-score in people with moderate to high cardiometabolic risk. However, from a pragmatic point of view, the issue of superiority might be less relevant since WB-EMS should be considered a training option predominately suitable for people with limited time resources, low affinity, or little motivation to exercise conventionally.

In reviewing the physiological mechanisms of MetS changes, the present work was unable to clarify which of the underlying parameters of MetS was most sensitive to WB-EMS. The results of MetS components of the individual trials (Table 4) indicate that only MAP shows consistently more favorable effects in the WB-EMS group compared with the CG, while all the other parameters (i.e., waist circumference, resting glucose, triglycerides, HDL-C) revealed inconsistent effects partially in favor of WB-EMS and partially in favor of the control. However, apart from the low number of studies, we abstained from sub-analyses of the five parameters constituting MetS according to IDF [9] or NCEP ATP III [10] because of the finding that particular laboratory biomarkers (i.e., FPG, HDL-C, TG) of most studies were in a normal range. Correspondingly, the clinical relevance of presumably low to moderate positive or negative changes will be difficult to estimate. Nevertheless, in reviewing the five studies, it was found that waist circumference decreased in all trials. Although the effects were not significant in each case, the clinical relevance of this aspect is important. In parallel, MAP significantly decreased in four of the five studies [13,14,22,24], while all trials reported at least suboptimum average baseline MAP (102–110 mmHg). Fasting glucose declined in all WB-EMS groups; however, because of the widely normal average baseline values (90 to 104 mg/dL) or/and minor changes, the results on fasting glucose should be considered clinically less relevant. The same is true for HDL-C with its either minor positive or minor negative changes (±2 mg/dL), a finding consistently observed by the five studies. In parallel, no study reported significant declines in triglyceride levels after WB-EMS.

Apart from the limited number of eligible studies and their low to moderate sample sizes, some other limitations and study particularities should be considered to interpret our findings reliably. (a) First of all, one may criticize that we did not include solely WB-EMS studies with non-training control groups. While the comparison of WB-EMS&HIIT versus isolated HIIT (and not non-training control) in the study by Amaro-Gahete et al. [14] is plausible and comprehensive, the inclusion of the study by Kemmler et al. [22] that compared isolated WB-EMS vs. isolated HIT-RT is more debatable. However, we finally decided to include the study bearing in mind that the comparison with a presumably effective intervention might dilute the actual effect of WB-EMS in the analysis. For this reason, we conducted a sensitivity analysis without the study by Kemmler et al. [22], which only slightly reduced the (albeit low) effect of WB-EMS on the MetS-Z-score.

(b) Another minor limitation is that this study is based on the comprehensive results of a systematic literature search (PRISMA) and evidence map of WB-EMS conducted up to March 2023. In order to check if eligible articles have been published after this date, we conducted an additional literature search (up to 28 September 2024) in electronic databases only. Furthermore, because our approach used the search of a previous comprehensive literature search, we are unable to fully apply the PRISMA criteria for the present article. In parallel, this study was not registered. (c) In order to include clinically relevant cohorts, we focused on cohorts with central, i.e., abdominal obesity. Actually, waist circumference (as the indicator of central obesity in all MetS definitions) is a valid determinant of intra-abdominal/visceral fat tissue accumulation [32], considered the key driver of cardiometabolic risk [33]. However, it should be noted that cut-off criteria for waist circumference differ considerably between the definitions decided by the IDF (≥80 and ≥94 cm) and ATP III (≥88 and ≥102 cm) for women and men, respectively. (d) The studies did not perform a homogeneous calculation of the MetS-Z-score: two studies each applied the NCEP ATP III cut-off values [13,23], and the three others used IDM criteria [14,22,24]. Additionally, while four studies properly applied the approach suggested by Johnson et al. [12], because of a lack of information, we cannot be sure if Amaro-Gehete et al. [14] strictly followed the specifications of Johnson et al. [12].

(e) We applied a random-effects meta-analysis with the inverse heterogeneity model (IVhet) [20], which is less susceptible to the underestimation of statistical error in heterogeneous studies. Therefore, the results are more reliable in heterogeneous studies, where the random effects estimator may lead to coverage probabilities that are well below the desired nominal level, which means the significance and relevance of the results may be overestimated [34]. Considering the low heterogeneity listed in Figure 1 (I^2^ = 0%), one may argue that a random effects meta-analysis is not appropriate. However, with only five eligible trials, we were not in a suitable position to prove heterogeneity statistically. We obtained an estimated I^2^ of 0%, but the confidence interval was very wide (0% to 77%) and thus did not even exclude “high/considerable heterogeneity”. Correspondingly, we were aware of higher degrees of heterogeneity and therefore applied a random effects model.

(f) All studies covered cohorts with central obesity, and two studies included people with MetS [13,23]. Thus, we think it is justified to generalize our findings to middle-aged to older people with increased cardiometabolic risk.

Considering the low withdrawal and high attendance rates of the WB-EMS study arms, WB-EMS can also be classified as an attractive training method. Furthermore, the dense network of commercial WB-EMS facilities, particularly in Germany [3], and its ongoing distribution worldwide indicate the feasibility and applicability of this novel training technology.

In summary, the present study suffers from large heterogeneity between the study protocols with respect to age, gender, training/non-training control groups, length of the intervention, and weekly WB-EMS training frequency. With respect to the stimulation protocol, all studies applied low-stimulation frequency WB-EMS with intermitted (predominately 4–6 s of impulse/4 s of impulse break) stimuli; nevertheless, the superimposed approach of Amaro-Gahete et al. [14] complicates the proper assignment of the effect and/or dilutes the difference in MetS-Z-score changes compared with a training control group.

Bearing the above in mind, we would like to conclude that we provided at least low evidence for a favorable effect of WB-EMS on the metabolic syndrome in cohorts at increased cardiometabolic risk. Apart from its effectiveness, WB-EMS can be considered a feasible attractive, and safe training option particularly suitable for people unable or unmotivated to exercise conventionally.

## Figures and Tables

**Figure 1 sensors-24-06788-f001:**
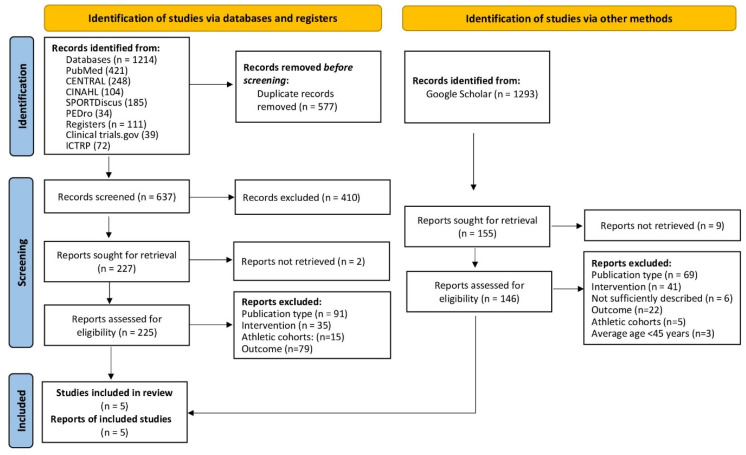
Flowchart of the present literature search according to PRISMA [25].

**Figure 2 sensors-24-06788-f002:**
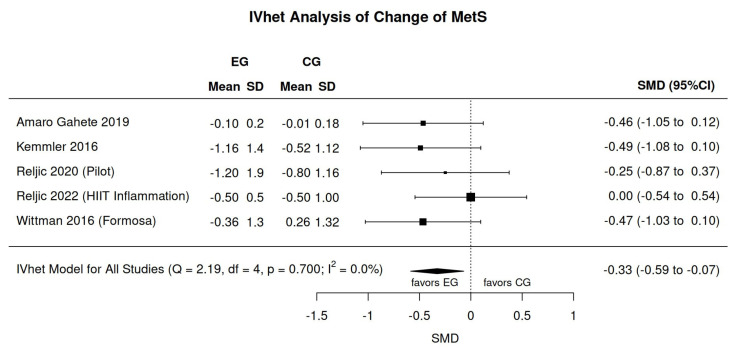
Forest plot showing the meta-analysis results of all the included trials [13,14,22,23,24] for WB-EMS effects on the metabolic syndrome score. Data are shown as pooled standard mean differences (SMDs) with 95%-CIs for changes after WB-EMS (EG) versus control (CG).

**Figure 3 sensors-24-06788-f003:**
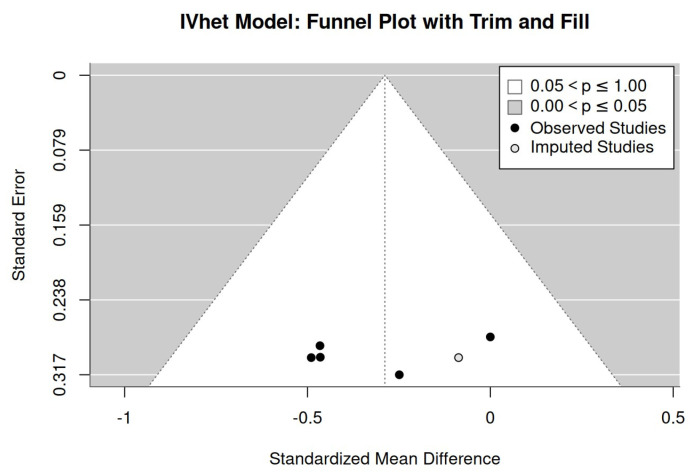
IV-Het funnel plot with trim-and-fill for WB-EMS effects on the metabolic syndrome score.

**Table 1 sensors-24-06788-t001:** Study, participant, and diet characteristics of the five trials included in the present systematic review and meta-analysis.

First Author, Year		Study Design	Sample Size/Group [n]	Gender (Men/Women)	Age [Years]	Body Mass Index [kg/m^2^]	Waist Circum- Ference (cm)	Dietary Intervention/Energy Restriction	Cardio- Vascular Health Status
1	Amaro-Gahete et al. 2019 [14]	RCT	WB-EMS: 19 CG: 18	WB-EMS:10/9 CG: 9/9	WB-EMS: 53.5 ± 5.3 CG: 53.1 ± 5.6	28.6 ± 4.6 26.4 ± 3.2	99.3 ± 13.7 97.5 ± 10.9	no	MR MR
2	Kemmler et al. 2016 [22]	RCT	WB-EMS: 23 CG: 23	Only men	WB-EMS: 43.7 ± 6.1 CG: 41.9 ± 6.4	28.5 ± 4.1 26.9 ± 3.3	102.6 ± 9.4 100.5 ± 9.6	no	MR
3	Reljic et al. 2020 [23]	RCT	WB-EMS: 15 CG: 14	Only women	56.0 ± 10.9 Details n.g.	36.1 ± 4.5 37.4 ± 4.8	107.2 ± 7.3 109.6 ± 8.6	−500 kcal/d + Protein ≥ 1 g/d −500 kcal/d+ Protein ≥ 1 g/d	HR
4	Reljic et al. 2022 [13]	RCT	WB-EMS: 26 CG: 26	WB-EMS: 8/18 CG: 8/18	WB-EMS: 52.7 ± 12.5 CG: 49.0 ± 15.1	37.2 ± 4.0 38.0 ± 6.3	114 ± 10 109 ± 11	−500 kcal/d/ −500 kcal/d	HR
5	Wittmann et al. 2016 [24]	RCT	WB-EMS: 25 CG: 25	Only women	WB-EMS: 77.3 ± 4.9 CG: 77.4 ± 4.9	24.2 ± 2.0 23.9 ± 1.4	93.5 ± 4.8 91.4 ± 6.4	no	MR

AE: aerobic training; MR: moderate cardiovascular risk (e.g., central obesity); HR: high cardiovascular risk (e.g., prevalent MetS), m: men; n.g. not given; RT: resistance training; w: women.

**Table 2 sensors-24-06788-t002:** Exercise characteristics of the five trials included in the present systematic review and meta-analysis.

First Author, Year		Superimposed WB-EMS	Intervention Length [Weeks]	Sessions/Week [n]	Length of Session [min]	Impulse Frequency [Hz]	Impulse Intensity	Duty Cycle [%] Impulse- Rest Phase	Control Physical Intervention	Loss to Follow-Up [%]	Attendance [%]	Adverse Effects
1	Amaro-Gahete et al. 2019 [14]	HIIT AE + RT and WB-EMS	12	2	20, 33	10–20, 35–75	moderate–high	AE: 99 RT: 50–63	HIIT AE + RT	HIIT + WB-EMS: 17 HIIT: 30	HIIT+WB-EMS: 99 HIIT: 99	no
2	Kemmler et al. 2016 [22]	no	16	1.5	20	85	high	60 6–4 s	HIT-RT	WB-EMS: 9 HIT-RT: 13	WB-EMS: 90 ± 11 HIT-RT: 93 ± 7	no
3	Reljic et al. 2020 [23]	no	12	2	20	85	moderate	60 6–4 s	none	25	93 ± 8	no
4	Reljic et al. 2022 [13]	no	12	2	20	85	moderate	60 6–4 s	none	23	93 ± 8	no
5	Wittmann et al. [24]	no	26	1	20	85	low–moderate	50 4–4 s	none	4	89 ± 6	no

AE: aerobic training; HIT-RT: single-set, high-intensity resistance exercise training, HIIT: high-intensity interval training, MR: moderate cardiovascular risk (e.g., central obesity); HR: high cardiovascular risk (e.g., prevalent MetS), m: men; RT: resistance training; w: women.

**Table 3 sensors-24-06788-t003:** Assessment of risk of bias for the included studies.

First Author, Year	Eligibility Criteria	Random Allocation	Allocation Concealment	Inter Group Homogeneity	Blinding Subjects	Blinding Personnel	Blinding Assessors	Participation ≥ 85% Allocation	Intention to Treat Analysis ^a^	Between Group Comparison	Measure of Variability	Total Score
Amaro-Gahete et al. 2019 [14]	Y	1	1	1	0	0	0	0	1	1	1	**6**
Kemmler et al. 2016 [22]	Y	1	0	1	0	0	1	1	1	1	1	**7**
Reljic et al. 2020 [23]	Y	1	1	1	0	0	1	0	0	1	1	**6**
Reljic et al. 2022 [13]	Y	1	1	1	0	0	1	0	0	1	1	**6**
Wittmann et al. 2016 [24]	Y	1	1	1	0	0	1	1	1	1	1	**8**

^a^ A point is awarded not only for the intention to treat analysis but also when “all subjects for whom outcome measures were available received the treatment or control condition as allocated”. Bold: Total PEDro-score.

**Table 4 sensors-24-06788-t004:** Changes in MetS-Score components in the WB-EMS and control groups of the five studies.

	Amaro-Gahete et al. 2019 [14] ^1^	Kemmler et al. 2016 [22]	Reljic et al. 2020 [23] ^2^	Reljic et al. 2022 [13] ^1,2^	Wittmann et al. 2016 [24]
Δ Waist circumference WB-EMS (cm)	−4.0 ± 2.4	−3.4 ± 4.5	−2.3	−3.0	−1.4 ± 2.1
Δ Waist circumference Control (cm)	−4.5 ± 2.5	−2.1 ± 4.1	−1.0	−2.0	−0.0 ± 2.3
Δ MAP WB-EMS (mmHg)	−5.4 ± 3.1	−4.9 ± 7.3	−7.0	2.0	**−8.8 ± 11.0**
Δ MAP Control (mmHg)	−1.6 ± 1.8	−3.6 ± 5.6	1.0	−1.0	**−2.2 ± 9.5**
Δ Triglycerides WB-EMS (mg/dL)	−30 ± 41	9.5 ± 55.5	−6.0	−15.0	2.8 ± 28.5
Δ Triglycerides Control (mg/dL)	−15 ± 60	−10.1 ± 47.9	−30.0	−18.0	9.8 ± 39.2
Δ HDL-C WB-EMS (mg/dL)	5.1 ± 12.9	n.g. ^3^	−1.0	−1.0	** *−1.3 ± 6.35* **
Δ HDL-C Control (mg/dL)	2.2 ± 12.8	n.g.	0	−2.0	** *−4.6 ± 6.6* **
Δ Fasting Glucose WB-EMS (mg/dL)	0.6 ± 5.9	−4.3 ± 9.0	−2.0	−2.0	−3.0 ± 10.3
Δ Fasting Glucose Control (mg/dL)	−4.1 ± 6.1	1.7 ± 8.5	−5.0	−3.0	−3.6 ± 7.9

Bold values: significant effects in favor of WB-EM; bold and italic: significant effects in favor of the control group; ^1^ significant differences between WB-EMS and CG were not calculated; ^2^ differences between pre- and post-intervention were not given and therefore calculated; ^3^ however, a significant effect for the total cholesterol/HDL-C ratio in favor of the WB-EMS group was not given.

## Data Availability

The datasets generated and/or analyzed during the current study are available from the corresponding author upon reasonable request.

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
