# Peer review of "Effects of Whole-Body Electromyostimulation on Metabolic Syndrome in Adults at Moderate-to-High Cardiometabolic Risk—A Systematic Review and Meta-Analysis"

_sensors, 2024, doi:10.3390/s24216788_

Round 1
Reviewer 1 Report
Comments and Suggestions for Authors
The paper represents a mini-review and meta-analysis of 5 controlled trials that examined the effect of whole-body electrostimulation (WB-EMS) on metabolic syndrome (MetS) in people with abdominal obesity, expressed as „the MetS-Z-Score“, calculated by the formula that has included all components of MetS, in comparison with no treatment or other physical activity treatments. Results were expressed as standardized mean differences (SMD) with 95%-confidence intervals (95%-CI), and a „low“ positive effect was shown (SMD: -0.30; 95%-CI: -0.04 to -0.56).
The topic is quite interesting, but many pitfalls exist.
1. Studies included: There was a small number of studies included, with a limited number of participants and with high heterogeneity regarding sex, age, body mass index, the METs definition criteria, control treatment, duration of treatment, the intensity of treatments, and the way the MetS Z score was calculated.
2. The most important is that one study [21] was a comparison with another physical activity treatment, while the other studies were a comparison with no treatment (4) or a comparison of a physical activity treatment with or without added WB-EMS (1). For those reasons, it is necessary to make additional calculations with the exclusion of the study in reference [21].
3. Data in Table 1 should be given per WB-EMS group and control group.
4. There are mistakes in the Table 1. For example, in [13] the age in the 2 examined groups (WB-EMS and control group) was 52.7 ± 12.5 and 49.0 ± 15.1, respectively, while in Table 1 stands 56±1 https://www.ncbi.nlm.nih.gov/pmc/articles/PMC9145085/ Please, check all data in table 1.
5. In that study, [13] there was a significant decrease in total caloric intake in the WB-EMS group (-500 kcal, by nutritional counseling, and the control group received the same counseling, but decreased by 400 kcal), which is very important. However, it was not mentioned in the manuscript that also nutritional caloric reduction by 500 kcal was included. What about other studies? Please, give also this information on dietary intervention.
6. There are mistakes in figure 2. E.g., in that study [13], the MetS score was changed completely the same as in the control treatment, -0.5 in both (figure 2 https://www.ncbi.nlm.nih.gov/pmc/articles/PMC9145085/), while according to your figure 2, the WB-EMS had -0.5 while the control had -0.7, what is not true.
7. In that study, [13], only WB-EMS and the control group did not have a positive effect on the MetS, even though there was a NS reduction). while all other physical activity treatments had (even in one of them it was NS). This should be pointed out in the narrative review.
8. Please, check the data from all studies included, and make again calculations. Be sure that there are no further mistakes.
9. There are missing very important additional tables (in the main manuscript or supplement) with information for quality assessment of studies and heterogeneity among studies that you included. Please, add them.
10. There is no rationale for the statement: „Nevertheless, the present study provided evidence that 200 WB-EMS is roughly as effective as conventional exercise in positively impacting cardiometabolic risk. (lines 200-201): your meta-analysis compared mostly with no treatment (4/5 studies). Please, eliminate this statement.
11. In table 1, define abbreviations HIIT AE and HIT
12. Abstract: define the abbreviation WB-EMS (whole-body electrostimulation)
13. In line 50, please, give more information on the MetS-49 (Z)-Score, and how it is calculated.
14. Methodology: Lines 163-165: Give information on how continuous MetS-49 (Z)-Score is calculated. Show calculations for MetS-49 (Z)-Score calculations for study [14] and for females (e.g., similarly as https://www.ncbi.nlm.nih.gov/pmc/articles/PMC9145085/). The formula should not be in the footnotes. please, eliminate footnotes, and describe properly.
15. lines 70-71: define how central obesity was defined (by measurements of waist circumference? Please, give the cut-offs)
16. line 107: Please, also state which software you used for analysis (if it is R you used? please, write it)
17. In lines 107-109, you stated that you use „random effects“ for analysis and that the heterogeneity model by Doi et al [19] was used. Then later on in lines 169-170, you state that heterogeneity was I2=0, very low (how that? The studies were heterogenous, is it obvious from figure 2 and Table 1?). Random effects for analysis were used in your study, but they should be used when I2 is higher than 50%, and if it is Doi's proposed alternative method, why did you state that you used random effect? Please, clarify: how you got such a low I2 (while the studies were obviously very heterogogenic, with high inter-study heterogeneity), and why when I2 was low, you used a random-effects model, while a fixed-effects model should be applied?).
https://meta-analysis.com/download/commonmistakes/Common%20Mistakes%20-%20Heterogeneity.pdf
https://www.um.es/metaanalysis/pdf/5008.pdf : „So, when the studies’ results only differ by the sampling error (homogeneous case), a fixed-effects model can be applied to obtain an average effect size. By contrast, if the study results differ by more than the sampling error (heterogeneous case), then the meta-analyst can assume a random-effects model, in order to take into account both within- and between-studies variability, or can decide to search for moderator variables from a fixed-effects model.”....
„The I2 index can be interpreted as the percentage of the total variability in a set of effect sizes due to true heterogeneity, that is, to between-studies variability. For example, a meta-analysis with I2 0 means that all variability in effect size estimates is due to sampling error within studies. On the other hand, a meta-analysis with I2 50 means that half of the total variability among effect sizes is caused not by sampling error but by true heterogeneity between studies. Higgins and Thompson proposed a tentative classification of I2 values with the purpose of helping to interpret its magnitude. Thus, percentages of around 25% (I2 25), 50% (I2 50), and 75% (I2 75) would mean low, medium, and high heterogeneity, respectively.“
18. Line 117. Should be: SMD values of >0.2, >0.5, and >0.8 were interpreted as small, medium, and large effects.
19. Lines 138-151 : It is obvious that heterogeneity is high. Please, give also information on dietary changes.
20. Line 139: What is meant by “All other studies focus on WB-EMS with low-intensity voluntary exercises/movements.” – It is confusing: 3 studies were compared with no treatment, not to physical activity, only 1 study [21] with exercise.
21. 162. Specify the cut-off differences for 2 definitions of MetS used.
22. Calculate also SMD without study 21
23. 169: Heterogeneity between the trials was very low (I2: 0%, Fig 2). – from figure 2 it is obvious that there was heterogeneity in the effect (study [13]). Please, correct the data from the studies, and calculate I2 the way it is proposed in: https://handbook-5-1.cochrane.org/chapter_9/9_5_2_identifying_and_measuring_heterogeneity.htm
![]()
24. Lines 191-221. Should go to Results (of narrative review), not Discussion.
25. 230-231: „Another limitation is that studies on WB-EMS and the MetS published after March 2023 were not considered. However, we are not aware of any such studies“- Please, check if there are such studies and preformulate the date.
26. 238-241: „We applied the inverse heterogeneity model (IVhet) [19] that is less susceptible to underestimation of statistical error in heterogeneous studies; i.e., the results are more reliable in heterogeneous studies especially with respect to the coverage probability of confidence intervals [31].“ – what is meant by that?
27. 244-250- (f) –this should be a separate paragraph, not in paragraph for study limitations.

no comments
Reviewer 2 Report
Comments and Suggestions for Authors
The study’s methods was well conducted, with a rigorous approach to data collection and analysis. However, the conclusion presented by the authors does not adequately account for some important limitations. Firstly, the small number of studies included, and consequently the reduced sample size of participants, affects the robustness of the results. Additionally, there is significant heterogeneity in the population analyzed, with participants ranging in age from 43 to 77 years and including both sexes, which may introduce variability into the results and make it harder to generalize the conclusions.
Another critical point is the variation in the intervention duration across studies, ranging from 12 to 24 weeks, with some studies lasting twice as long as others. This difference could have a considerable impact on the magnitude of the observed effects, making comparisons less consistent. It is also important to highlight that there were interventions combined with other treatments, which the authors acknowledge in the limitations but should have emphasized more in the interpretation of the results.
Therefore, when stating, "In summary, we provided evidence for the favorable effect of WB-EMS on the 'Metabolic Syndrome' in cohorts at increased cardiometabolic risk," the authors seem to overlook the complexity of the data and the need for a more cautious interpretation.
Additionally, it would be beneficial to include in the paper a table or chart summarizing the articles and indicating where the changes occurred, considering the criteria for Metabolic Syndrome. This would provide a clearer overview of the key findings, allow for easier comparison across studies, and help highlight the specific interventions and their effects on the components of Metabolic Syndrome. Such a summary could enhance the clarity of the analysis and provide a visual aid to better understand the heterogeneity in the results and the various factors at play.
Round 2
Reviewer 1 Report
Comments and Suggestions for Authors
The manuscript was substantially improved.